SciPost Physics                                        

# Collective Radiative Interactions in the Discrete Truncated Wigner Approximation

Christopher D. Mink and Michael Fleischhauer

University of Kaiserslautern-Landau
\* cmink@rptu.de

November 14, 2023

## Abstract

Interfaces of light and matter serve as a platform for exciting many-body physics and photonic quantum technologies. Due to the recent experimental realization of atomic arrays at sub-wavelength spacings, collective interaction effects such as superradiance have regained substantial interest. Their analytical and numerical treatment is however quite challenging. Here we develop a semiclassical approach to this problem that allows to describe the coherent and dissipative many-body dynamics of interacting spins while taking into account lowest-order quantum fluctuations. For this purpose we extend the discrete truncated Wigner approximation, originally developed for unitarily coupled spins, to include collective, dissipative spin processes by means of truncated correspondence rules. This maps the dynamics of the atomic ensemble onto a set of semiclassical, numerically inexpensive stochastic differential equations. We benchmark our method with exact results for the case of Dicke decay, which shows excellent agreement. For small arrays we compare to exact simulations and a second order cumulant expansion, again showing good agreement at early times and at moderate to strong driving. We conclude by studying the radiative properties of a spatially extended three-dimensional, coherently driven gas and compare the coherence of the emitted light to experimental results.

# 1 Introduction

The accurate description of non-equilibrium dynamics of interacting quantum spin systems is one of the major challenges of many-body theory. At the same time it is of central importance in many areas of physics. A prime example is the collective interaction of two-level atoms with the quantized electromagnetic field, which after integrating out the radiation field can be mapped onto collectively coupled spins with long-range interactions and dissipation. Collective light-matter interactions have been a central problem in quantum optics starting from the early work of Dicke [1]. Dicke showed that an ensemble of closely spaced two-level quantum emitters can display intriguing collective effects in the emission of light such as sub- and superradiance [2, 3] observed in a number of experiments [4–6]. This collective coupling between light and atoms has recently regained substantial interest as it is at the heart of many photonic quantum technologies [7]. Collective light-atom couplings are for example the basis of ensemble-based quantum memories for photons [8–10], quantum repeaters [11], and many concepts for realizing strongly interacting photons [12–15]. Here the interplay of the nonlinear atomic response and quantum entanglement results in rich coherent many-body dynamics.

    A comprehensive theoretical treatment of the collective interaction of light with quantum emitters is however only simple if the spatial extension of the emitters can be neglected as in the case of the Dicke model or in cavity QED. Spatially extended systems can only be described by solving the master equation, e.g. by Monte-Carlo Wave Function (MCWF) simulations [16], if the number of excitations is small or for small ensemble sizes. Numerical techniques based on matrix product states [17], which have proven to be extremely powerful for one-dimensional systems with short-range couplings are usually not appropriate in higher spatial dimensions and for long-range couplings. Likewise a classical treatment of collective phenomena in terms of Maxwell-Bloch equations does not capture the buildup of quantum correlations between the atoms. While some universal features of superradiance can be predicted for spatially extended systems without involved numerics [18, 19], there is for example no simple access to the timing and intensity of superradiant bursts. Expanding on the classical mean-field description in terms of Maxwell-Bloch equations, cumulant expansion techniques have been employed to account for correlations [20–22], but generally require higher order expansions for accurate predictions. Cumulant expansions are furthermore often ill-controlled and can suffer from intrinsic instabilities. Moreover their numerical complexity grows as a power law of the expansion order $n$, i.e. scales as $N^n$, where $N$ is the number of spins, making them computationally expensive. The same holds for non-equilibrium Greens function approaches such as the one employed in [23].

    In the present paper we propose an alternative, semiclassical approach that allows to describe the coherent and dissipative many-body dynamics of interacting spins, taking into ac-

count lowest-order quantum fluctuations. Our approach is inspired by the success of the discrete truncated Wigner approximation (DTWA) for the treatment of unitarily interacting spin systems [24], which has recently been extended to include single-particle dissipation [25–27]. Within the truncated Wigner approximation the dissipative many-body dynamics of spins is mapped to a generalized diffusion problem of the Wigner quasi-probability distribution in phase space. The exact relation between the dynamics of the many-body density matrix in Hilbert space and the Wigner function in phase space is given by correspondence rules, which lead to higher-order partial differential equations for the Wigner function. These are in general intractable without further approximations. A very successful approximation applicable to unitarily coupled spins is the DTWA, which can be extended to include single particle decay and dephasing [25]. The approach of Ref. [25] is however not useful for collective dissipative processes such as superradiance. We here pursue a different route. We propose approximate correspondence rules which lead to Fokker-Planck type equations of motion for the Wigner quasi distribution equivalent to a set of coupled stochastic differential equations (SDEs) for the spin orientations. Since the number of these equations scales linearly in the number of spins, the solution is numerically inexpensive and allows investigating system sizes much larger than in other semiclassical approaches. In the truncated Wigner approximation quantum fluctuations are taken into account to lowest order by nondeterministic initial conditions and by collectively coupling the spins to white noise processes, which generate (weak) entanglement between the spins.

Our paper is organized as follows: In Sect. 2 we give a compact summary of the continuous Wigner phase space representation of an ensemble of two-level systems (spins). We introduce a truncated Wigner Approximation for spin ensembles with *collective* couplings in Sect. 3. In particular we propose and motivate approximate correspondence rules and discuss general conditions for their validity. The main application of our methods are collective light-matter couplings in free space, which we will introduce in Sect. 4. In Sect. 5 we benchmark our method for the Dicke decay, i.e. the collective emission of light from a tightly localized ensemble of two-level atoms, for which the full master equation can be solved exactly. We find excellent agreement and give a physical interpretation of the emerging collective response within the semiclassical approximation. We then study collective light-matter phenomena in spatially extended systems in Sect. 6, where the full dynamics can no longer be described exactly. We consider superradiance from an elongated cloud of coherently driven atoms as well as regular arrays of atoms. Finally Sect. 7 summarizes the results and gives an outlook to future work.

## 2 Wigner Representation for Spins

An approach widely used in quantum optics to describe the dynamics of interacting, driven-dissipative many-body systems beyond the mean-field level is the truncated Wigner approximation (TWA) [28–32]. It describes interactions on a mean-field level but allows taking both thermal and leading-order quantum fluctuations into account by averaging over nondeterministic initial conditions and by coupling to stochastic noise sources. In the following we will give a compact summary of the Wigner representation of an ensemble of two-level systems or spins, but refer to Refs. [33, 34] for a more general introduction to phase-space representations in quantum mechanics. We formulate the TWA by studying the correspondence rules [35], which translate the action of an operator in Hilbert space to a differential operator in phase space, and show that they have a simple asymptotic limit for collective processes.

## 2.1 Wigner representation of two-level systems

The connection between Hilbert space and Wigner phase space, spanned by some c-number variables $\mathbf{\Omega}$ is given by the representation of an operator $\hat{O}$ in terms of a complex function $W_{\hat{O}}(\mathbf{\Omega})$, called the Weyl symbol

$$\hat{O} = \int d\mathbf{\Omega}\, W_{\hat{O}}(\mathbf{\Omega})\,\hat{\Delta}(\mathbf{\Omega}). \tag{1}$$

Here $\hat{\Delta}(\mathbf{\Omega})$ is the so-called phase point operator or Wigner kernel. Inversely the Weyl symbol can be expressed explicitly in terms of the operator by

$$W_{\hat{O}}(\mathbf{\Omega}) = \mathrm{Tr}\Big[\hat{\Delta}(\mathbf{\Omega})\,\hat{O}\Big]. \tag{2}$$

Of particular interest is the Weyl symbol $W_{\hat{\rho}}(\mathbf{\Omega})$ of the density operator $\hat{\rho}$, which is called the Wigner function or Wigner (quasi-probability) distribution. The latter notion is due to the fact that $W_{\hat{\rho}}(\mathbf{\Omega}) \in \mathbb{R}$ and

$$\int d\mathbf{\Omega}\, W_{\hat{\rho}}(\mathbf{\Omega}) = \mathrm{Tr}(\hat{\rho}) = 1, \tag{3}$$

but can be negative.

Originally formulated for continuous degrees of freedom the concept of phase space representations can be extended to systems with finite-dimensional Hilbert spaces [36] such as spin-$\frac{1}{2}$ systems. There is however some freedom in choosing the phase point operators. A specific discrete representation has been introduced by Wooters in [36], which is the foundation of the discrete truncated Wigner approximation [24]. Here we adopt however a different, continuous representation of spin-1/2 states $\hat{\rho}$ through rotations of the particular discrete phase point operator $\hat{\Delta}_0 = \frac{1}{2}(\hat{\mathbb{1}}_2 + \sqrt{3}\hat{\sigma}^z)$

$$\hat{\Delta}(\theta, \phi) = U(\theta, \phi, \psi)\hat{\Delta}_0 U^{\dagger}(\theta, \phi, \psi), \quad \Omega = (\theta, \phi) \tag{4}$$

which was shown in [25] to be more appropriate to describe dissipative spin systems. Here $U(\theta, \phi, \psi) = e^{-i\hat{\sigma}^z\phi/2}e^{-i\hat{\sigma}^y\theta/2}e^{-i\hat{\sigma}^z\psi/2}$, are the SU(2) rotation operators with Euler angles $(\theta, \phi, \psi)$, which gives

$$\hat{\Delta}(\theta, \phi) = \frac{1}{2}\Big[\hat{\mathbb{1}}_2 + \boldsymbol{s}(\theta, \phi)\hat{\boldsymbol{\sigma}}\Big] = \frac{1}{2}\begin{pmatrix} 1 + \sqrt{3}\cos\theta & \sqrt{3}e^{-i\phi}\sin\theta \\ \sqrt{3}e^{i\phi}\sin\theta & 1 - \sqrt{3}\cos\theta \end{pmatrix}, \tag{5}$$

Note that in [25] we have used a slightly different definition of the Wigner kernel that is obtained by letting $\theta \to \pi - \theta$ and $\phi \to -\phi$. Based on the same work, we have shown that, using a gauge freedom, the most relevant states $|\downarrow\rangle$ and $|\uparrow\rangle$ can be represented by simple, positive Wigner functions. Namely

$$W_{|\psi\rangle\langle\psi|}(\theta, \phi) = \frac{1}{\sin\theta_{\psi}}\delta(\theta_{\psi} - \theta), \tag{6a}$$

$$\theta_{\psi} = \begin{cases} \arccos\left(\frac{1}{\sqrt{3}}\right), & \psi = \uparrow \\ \pi - \arccos\left(\frac{1}{\sqrt{3}}\right), & \psi = \downarrow \end{cases}, \tag{6b}$$

which are straightforwardly verified by substituting them into Eq. (1). The above Wigner function can be sampled by a fixed $\theta = \theta_{\psi}$ and drawing $0 \le \phi < 2\pi$ from a uniformly random

distribution. We note furthermore that the vector $s(\theta, \phi)$ appearing in Eq. (5) is just the Weyl symbol of the Pauli spin-matrices

$$s(\theta, \phi) \equiv W_{\hat{\sigma}}(\theta, \phi) = \sqrt{3}\left(\sin\theta\cos\phi, \sin\theta\sin\phi, \cos\theta\right)^T. \tag{7}$$

Similarly, the Weyl symbols of the creation- and annihilation operators $\hat{\sigma}^{\pm} = (\hat{\sigma}^x \pm i\hat{\sigma}^y)/2$ are given by

$$s^{\pm}(\Omega) \equiv W_{\hat{\sigma}^{\pm}}(\Omega) = \frac{\sqrt{3}}{2}\sin\theta\, e^{i\phi}. \tag{8}$$

The kernel in Eq. (5) can easily be extended to a system of $N$ spin-1/2 systems via $s \to s_n$ and $\Omega \to \mathbf{\Omega} = \{\Omega_n\}$, with $j = 1, 2, \ldots, N$ labelling the spins.

## 2.2  Time evolution of the Wigner function

Our goal is to find an approximate solution of the master equation of many-body spin systems

$$\frac{d}{dt}\hat{\rho} = -i[\hat{H}, \hat{\rho}] + \frac{1}{2}\sum_{\mu}\left(2\hat{L}_{\mu}\hat{\rho}\hat{L}_{\mu}^{\dagger} - \{\hat{L}_{\mu}^{\dagger}\hat{L}_{\mu}, \hat{\rho}\}\right) \tag{9}$$

where the many-body Hamiltonian $\hat{H}$ and the Lindblad operators $\hat{L}_{\mu}$, describing Markovian dissipative processes, are some functions of the spin-1/2 operators $\hat{\sigma}_j^{\mu}$. Generically $\hat{H}$ and/or the $\hat{L}_{\mu}$ describe interactions between spins which are higher dimensional, i.e. have couplings that cannot be reduced to a one-dimensional topology. The latter excludes in general efficient descriptions in terms of matrix product states [17].

To develop an approximate, semiclassical approach we need to translate the master equation of the density operator $\hat{\rho}$ into an equation of motion for the Wigner function $W_{\hat{\rho}}(\mathbf{\Omega})$. As the terms on the right hand side of Eq. (9) can be decomposed into products of spin operators and the density operator, this requires expressing the Weyl symbol of a composition of operators as emerging on the r.h.s. of (9), e.g. $\hat{H}\hat{\rho} \to W_{\hat{H}\hat{\rho}}$ in terms of the individual symbols $W_{\hat{H}}$ and $W_{\hat{\rho}}$. In phase space the Weyl symbol of a product does not correspond to a simple multiplication $W_{\hat{A}\hat{B}} \neq W_{\hat{A}} \cdot W_{\hat{B}}$ of the scalar functions. Instead, the composition is given by the *Moyal product* or *star product*

$$W_{\hat{A}\hat{B}}(\mathbf{\Omega}) = W_{\hat{A}} \star W_{\hat{B}} = \iint d\mathbf{\Omega}' d\mathbf{\Omega}'' \, W_{\hat{A}}(\mathbf{\Omega}') W_{\hat{B}}(\mathbf{\Omega}'') \, \mathrm{Tr}\left[\hat{\Delta}(\mathbf{\Omega})\hat{\Delta}(\mathbf{\Omega}')\hat{\Delta}(\mathbf{\Omega}'')\right], \tag{10}$$

which also has a differential form [33, 37]. The so called *correspondence rules* allow us to express the star product of Weyl symbols involving a spin operator, such as $W_{\hat{\sigma}_j^{\mu}} \star W_{\hat{A}}$, as differential operators acting on $W_{\hat{A}}$. With these rules we can iteratively translate compositions of operators as they appear in the master equation of $\hat{\rho}$ into a partial differential equation for the Wigner function $W_{\hat{\rho}}$.

A more direct approach for deriving phase space equations that we have recently considered [25] is based on a simple observation: For a continuous phase space representation of a single spin, generated by the kernel given in Eq. (5), the matrices $\hat{\Delta}, \partial_{\theta}\hat{\Delta}, \partial_{\phi}\hat{\Delta}$ and $\partial_{\phi}^2\hat{\Delta}$ span the Hilbert space. Hence any product of operators $\hat{O}\hat{\Delta}$ or $\hat{\Delta}\hat{O}$ can be expressed as a differential operator acting on $\hat{\Delta}$. Therefore, as can be seen from Eq. (1), the same operator acting on $\hat{\rho}$ can be converted into a differential operator acting on the Wigner function. However, the infinitesimal volume elements $d\Omega_n$ are not constant due to the curved phase space. It is therefore instructive to express the correspondence rules in terms of the contravariant coordinates

$(x^1, x^2) = (\theta, \phi)$ of the phase space of a single spin and the metric tensor $g_{\mu\nu}$ which is given by

$$g = \frac{1}{2\pi} \begin{pmatrix} 1 & 0 \\ 0 & \sin^2 \theta \end{pmatrix}. \tag{11}$$

The derivatives acting on the Wigner function are then given by covariant derivatives

$$\nabla_{x_n} = \frac{1}{\sqrt{\det(g)}} \frac{\partial}{\partial x_n} \sqrt{\det(g)} = \csc \theta_n \frac{\partial}{\partial x_n} \sin \theta_n \tag{12}$$

with $x_n = \theta_n, \phi_n$. This yields correspondence rules such as [25]

$$\hat{\sigma}^z \hat{\rho} \leftrightarrow \left[ \sqrt{3} \cos \theta + \nabla_\theta \frac{3 \sin \theta - 2 \csc \theta}{\sqrt{3}} - \nabla_\phi i + \nabla_\phi^2 \frac{2 \cot \theta \csc \theta}{\sqrt{3}} \right] W_{\hat{\rho}}(\Omega). \tag{13}$$

A full list of these rules, but in the aforementioned different angle convention, is given in Ref. [25].

For general spin-$j$ systems, exact expressions for the correspondence rules are known [35], but are complicated. They do have a simple semiclassical form in the limit $j \to \infty$, but for $j = 1/2$, which is by far the most commonly considered case in many branches of physics, this semiclassical limit is not directly applicable.

## 2.3 An example for an exact FPE: spontaneous emission of a single two-level atom

Let us start by applying the correspondence rules such as Eq. (13) to the important simple example of spontaneous decay, where an exact FPE can be derived. Two-level atoms in free space can undergo spontaneous relaxation to the energetically lower state by emission of light quanta. This is due to the fundamental coupling of the atoms to the quantized electromagnetic field. The description of this phenomenon can be drastically simplified by assuming that the field is in equilibrium (which is the vacuum at optical frequencies) and by subsequently integrating out the field's degrees of freedom. A Born-Markov approximation then yields the effective Lindblad master equation [38]

$$\frac{d}{dt} \hat{\rho} = \frac{\Gamma_0}{2} (2\hat{\sigma}^- \hat{\rho} \hat{\sigma}^+ - \hat{\sigma}^+ \hat{\sigma}^- \hat{\rho} - \hat{\rho} \hat{\sigma}^+ \hat{\sigma}^-) \tag{14}$$

for the density operator $\hat{\rho}(t)$ of an individual atom. The rate $\Gamma_0$ is the Einstein A coefficient.

Following the arguments of [25] we translate the master equation for $\hat{\rho}$ into an Fokker-Planck equation for the Wigner function $W_{\hat{\rho}}(\Omega, t)$:

$$\frac{\partial}{\partial t} W_{\hat{\rho}}(\Omega, t) = -\Gamma_0 \nabla_\theta \left( \cot \theta + \frac{\csc \theta}{\sqrt{3}} \right) W(\Omega, t)$$
$$+ \frac{\Gamma_0}{2} \nabla_\phi^2 \left( 1 + 2 \cot^2 \theta + \frac{2 \cot \theta \csc \theta}{\sqrt{3}} \right) W_{\hat{\rho}}(\Omega, t). \tag{15}$$

It has an equivalent set of Itô SDEs [39, 40]

$$d\theta = \Gamma_0 \left( \cot \theta + \frac{\csc \theta}{\sqrt{3}} \right) dt, \tag{16a}$$

$$d\phi = \sqrt{\Gamma_0 \left( 1 + 2 \cot^2 \theta + \frac{2 \cot \theta \csc \theta}{\sqrt{3}} \right)} dW_\phi, \tag{16b}$$

where $dW_\phi$ is the differential of a Wiener process. These equations are exact and solving them is numerically inexpensive without further approximation.

### 2.4 The general case: Truncated Wigner Approximations (TWA) as diffusion approximations

Knowing the exact phase space formulation of the master equation shifts the quantum many-body problem of solving a large matrix differential equation of the density operator $\hat{\rho}(t)$, Eq. (9), to solving a high-dimensional partial differential equation with possibly infinitely many orders of derivatives for the c-number quasi-distribution $W_{\hat{\rho}}(t)$. Except for special cases, such as the one discussed in Sec. 2.3, both formulations are useless without the introduction of further approximations.

From the perspective of complexity, the core idea behind different variants of the TWA consists of neglecting higher order terms of the equation of motion of $W_{\hat{\rho}}(\boldsymbol{\Omega}, t)$ such that the remaining expression is a covariant Fokker-Planck equation (FPE) in terms of suitable phase space variables $\boldsymbol{\Omega}$

$$\frac{\partial}{\partial t} W_{\hat{\rho}}(\boldsymbol{\Omega}, t) = -\sum_{x \in \boldsymbol{\Omega}} \nabla_x A_x(\boldsymbol{\Omega}, t) W_{\hat{\rho}}(\boldsymbol{\Omega}, t) + \frac{1}{2} \sum_{x,y \in \boldsymbol{\Omega}} \nabla_x \nabla_y D_{xy}(\boldsymbol{\Omega}, t) W_{\hat{\rho}}(\boldsymbol{\Omega}, t), \qquad (17)$$

where $\boldsymbol{D}(\boldsymbol{\Omega}, t) = \boldsymbol{B}(\boldsymbol{\Omega}, t)\boldsymbol{B}^T(\boldsymbol{\Omega}, t) \in \mathbb{R}^{2N \times 2N}$ is a positive semidefinite matrix. It then can be equivalently expressed by the set of SDEs [41]

$$d\boldsymbol{x} = \boldsymbol{A}(\boldsymbol{\Omega}, t)dt + \boldsymbol{B}(\boldsymbol{\Omega}, t)d\boldsymbol{W}, \qquad (18)$$

where $d\boldsymbol{W} \in \mathbb{R}^{2N}$ is a multivariate differential Wiener process. This and all further SDEs will implicitly be stated in the Itô calculus.

In a numerical implementation, we can efficiently compute $N_{\text{traj}}$ independent solutions of the SDEs [42], which we call *trajectories*. All relevant expectation values can then be directly calculated in the Wigner phase by using the relation

$$\text{Tr}\left(\hat{\rho}\hat{O}\right) = \int d\boldsymbol{\Omega} \, W_{\hat{\rho}}(\boldsymbol{\Omega}) W_{\hat{O}}(\boldsymbol{\Omega}) = \overline{W_{\hat{O}}(\boldsymbol{\Omega})}. \qquad (19)$$

The bars indicate the stochastic average

$$\overline{W_{\hat{O}}(\boldsymbol{\Omega})} \approx \frac{1}{N_{\text{traj}}} \sum_{n=1}^{N_{\text{traj}}} W_{\hat{O}}(\boldsymbol{\Omega}^{(n)}), \qquad (20)$$

where $\boldsymbol{\Omega}^{(n)}$ refers to the phase space coordinate of the $n$'th trajectory and where the approximation due to a stochastic error vanishes as we let $N_{\text{traj}} \to \infty$.

In summary, this means that the time evolution of the Wigner function in TWA is governed by a diffusion process in the spherical Wigner phase space. From a physical standpoint, this truncation should be formulated in a systematic fashion which elucidates its validity in terms of a small parameter.

When adding spin-spin interactions, such as Ising-type couplings, the resulting equation for $W_{\hat{\rho}}(\boldsymbol{\Omega}, t)$ is no longer of Fokker-Planck type and approximations are needed. A commonly used approximation is the discrete truncated Wigner approximation (DTWA) [24], which essentially amounts to a mean-field factorization of the Wigner function. This approach always produces deterministic equations and cannot account for the noise expected in dissipative systems.

As shown in [26, 43] independent dephasing of spins can be incorporated in the DTWA, but the description of decay requires some ad-hoc modelling [44], which is not justified in general. Therefore it is not surprising that the standard DTWA cannot be applied to collective decay processes such as superradiance. We recently developed an alternative approach, termed hybrid continuous-discrete truncated Wigner approximation (CDTWA), which describes the time

evolution of the many-body density operator by a *continuous* representation of the many-body spin Wigner function but samples the initial distribution from a discrete representation [25]. The CDTWA incorporates (uncorrelated) decay and dephasing of the spins in a consistent way, but in the context of collective decay does not generally reveal which correlated terms can be neglected or not (see Sec. IV. E of Ref. [25]).

## 3 Truncated Wigner Approximation for Large Spin Ensembles with Collective Couplings

Neither the standard DTWA nor the CDTWA mentioned in the previous section are suitable for describing problems of collective couplings among spins. We now present an alternative approach based on an approximate form of the correspondence rules for collective spin processes and derive conditions for their validity.

### 3.1 Semiclassical limit of the correspondence rules for collective operators

If an ensemble of two-level atoms is confined to a small volume comparable in size with the wavelength of the dipole transition between the two states, the coupling to the quantized electromagnetic field leads to a correlated emission of photons known as sub- and superradiance. Collective processes in an ensemble of $N$ spins can be described in terms of collective operators

$$\hat{S}(\{c_n\}) = \sum_{n=1}^{N} c_n \hat{\sigma}_n \tag{21}$$

where the "degree of cooperativity" is encoded in the weights $\{c_n\} = (c_1, c_2, \dots) \in \mathbb{C}^N$. For $c_1 = c_2 = \cdots = c_N$ the operator $\hat{S}(\{c_n\})$ describes the maximally cooperative case of an all-to-all coupling, relevant e.g. for modelling Dicke superradiance, see Sect. 4, while a distribution of the $c_n$'s peaked for some index $n = j$ corresponds to the low-cooperativity case of short-range interaction. The action of $\hat{S}(\{c_n\})$ on the state $\hat{\rho}$ can be exactly expressed as a differential operator acting on the Wigner function $W_{\hat{\rho}}(\Omega)$ in the phase space, however this differential operator does not have a simple form [34, 35]. For the resulting equation of motion for the Wigner function to be of practical use, we propose instead truncated correspondence rules

$$W_{\hat{S}(\{c_n\})\hat{\rho}}(\Omega) \approx \mathcal{S}(\{c_n\})W_{\hat{\rho}}(\Omega) = \sum_{n=1}^{N} c_n \left( s_n + L_n \right) W_{\hat{\rho}}(\Omega), \tag{22a}$$

$$\text{with} \quad L_n = i\nabla_{\theta_n} \begin{pmatrix} +\sin\phi_n \\ -\cos\phi_n \\ 0 \end{pmatrix} + i\nabla_{\phi_n} \begin{pmatrix} \cot\theta_n \cos\phi_n \\ \cot\theta_n \sin\phi_n \\ -1 \end{pmatrix}, \tag{22b}$$

where $s_n(\Omega)$ is given by Eq. (7) and $L_n$ is the angular momentum differential operator expressed in terms of covariant derivatives. Similarly we find the action

$$W_{\hat{\rho}\hat{S}(\{c_n\})}(\Omega) \approx \sum_{n=1}^{N} c_n \left( s_n - L_n \right) W_{\hat{\rho}}(\Omega) \tag{23}$$

for operators acting from the right-hand side. Note that the above contributions are the first and third term on the right-hand side of Eq. (13), whereas the remaining terms denote "quantum corrections" to this lowest order contribution. The intuition behind this truncation is simple: For a single spin-$j$, the same semiclassical limit can be obtained by letting $j \to \infty$. This reveals that classical and quantum contributions separate in the Wigner phase space.

We note that this approximation leads to a Fokker-Planck equation for $W_{\hat{\rho}}$ without higher-order derivatives if the master equation is at most bilinear in the collective operators. This allows for an efficient simulation in terms of SDEs. Eqs. (22) are the central element of our approach and form the basis of the simulations of collective decay phenomena discussed in Sects. 4 and 6.

## 3.2 Validity of the approximate correspondence rules

We now discuss the range of validity of the truncated correspondence rules, Eqs. (22). To this end we first note that the density operator $\hat{\rho}$ of a system of $N$ spins has the general form

$$\hat{\rho} = \sum_{\boldsymbol{\mu}} \rho_{\boldsymbol{\mu}} \hat{\sigma}_1^{\mu_1} \ldots \hat{\sigma}_N^{\mu_N}, \qquad \text{with} \quad \boldsymbol{\mu} = (\mu_1, \mu_2, \ldots), \tag{24}$$

and $\mu_n = (0, x, y, z)$, with $\hat{\sigma}^0 = \hat{\mathbb{1}}$ and $s^0 = 1$, from which we can immediately deduce

$$W_{\hat{\rho}}(\boldsymbol{\Omega}) = \sum_{\boldsymbol{\mu}} \rho_{\boldsymbol{\mu}} s_1^{\mu_1} \ldots s_N^{\mu_N}. \tag{25}$$

Note that this expression, while being exact, is only of formal use as the sum contains an exponentially large number of terms. It does allow us, however, to explicitly calculate the exact Weyl symbol of operators such as $\hat{S}^z(\{c_n\})\hat{\rho}$ through direct evaluation of Eq. (2) via Eq. (25):

$$W_{\hat{S}^z(\{c_n\})\hat{\rho}}(\boldsymbol{\Omega}) = \mathrm{Tr}\Big[\hat{\Delta}(\boldsymbol{\Omega})\hat{S}^z(\{c_n\})\hat{\rho}\Big] = \sum_{\boldsymbol{\mu}} \rho_{\boldsymbol{\mu}} \sum_n c_n \mathrm{Tr}\Big[\hat{\Delta}(\boldsymbol{\Omega})\hat{\sigma}_n^z \hat{\sigma}_1^{\mu_1} \ldots \hat{\sigma}_N^{\mu_N}\Big].$$

Applying the spin algebra of the Pauli matrices and evaluating the individual Weyl symbols yields

$$W_{\hat{S}^z(\{c_n\})\hat{\rho}}(\boldsymbol{\Omega}) = \sum_{\boldsymbol{\mu}} \rho_{\boldsymbol{\mu}} \sum_n s_1^{\mu_1} \ldots c_n \left(\delta_{\mu_n,0} s_n^z + \delta_{\mu_n,z} + i\varepsilon_{z,\mu_n,\nu_n} s_n^{\nu_n}\right) \ldots s_N^{\mu_N}, \tag{26}$$

where $\varepsilon_{ijk}$ is the Levi-Civita symbol. The truncation approximation in Eqs. (22) of the same Weyl symbol is obtained, on the other hand, by applying the $z$-component of Eq. (22a) to the Wigner function in Eq. (25), which yields:

$$\mathcal{S}^z(\{c_n\})W_{\hat{\rho}}(\boldsymbol{\Omega}) = \sum_{\boldsymbol{\mu}} \rho_{\boldsymbol{\mu}} \sum_n s_1^{\mu_1} \ldots c_n \left(s_n^z s_n^{\mu_n} + i\epsilon_{z,\mu_n,\nu_n} s_n^{\nu_n}\right) \ldots s_N^{\mu_N}. \tag{27}$$

To determine the error of Eq. (27) made by the truncated correspondence rule we define its difference to Eq. (26)

$$\begin{aligned} \delta^z(\boldsymbol{\Omega}) &\equiv W_{\hat{S}^z(\{c_n\})\hat{\rho}}(\boldsymbol{\Omega}) - \mathcal{S}^z(\{c_n\})W_{\hat{\rho}}(\boldsymbol{\Omega}) \\ &= \sum_{\boldsymbol{\mu}} \rho_{\boldsymbol{\mu}} \sum_n s_1^{\mu_1} \ldots c_n \left(\delta_{\mu_n,0} s_n^z + \delta_{\mu_n,z} - s_n^{\mu_n} s_n^z\right) \ldots s_N^{\mu_N} \\ &= \sum_{\boldsymbol{\mu}} \rho_{\boldsymbol{\mu}} \sum_n s_1^{\mu_1} \ldots c_n \left(1 - \delta_{\mu_n,0}\right)\left(\delta_{\mu_n,z} - s_n^{\mu_n} s_n^z\right) \ldots s_N^{\mu_N}. \end{aligned} \tag{28}$$

In a similar way we can proceed with the $x$- and $y$-components $\hat{S}^x(\{c_n\})\hat{\rho}$ and $\hat{S}^y(\{c_n\})\hat{\rho}$. This gives the full difference vector

$$\boldsymbol{\delta}(\boldsymbol{\Omega}) \equiv (\delta^x, \delta^y, \delta^z)^T(\boldsymbol{\Omega}) = \sum_{\boldsymbol{\mu}} \rho_{\boldsymbol{\mu}} \sum_n s_1^{\mu_1} \ldots c_n \left(1 - \delta_{\mu_n,0}\right)\left(\mathbb{1} - \boldsymbol{s}_n \boldsymbol{s}_n^T\right)^{\mu_n} \ldots s_N^{\mu_N}, \tag{29}$$

where the superscript $\mu_n$ indicates the $\mu_n$'th row of the given matrix. Finally, the error can be quantified by the norm of the vector $\delta(\Omega)$

$$
\begin{aligned}
||\delta(\Omega)||^2 &= \int d\Omega \left( |\delta^x|^2 + |\delta^y|^2 + |\delta^z|^2 \right) \\
&= \sum_{\mu,\nu} \rho_\mu^* \rho_\nu \sum_n \left( f_{nn}^{\mu\nu} + \sum_{m\neq n} f_{mn}^{\mu\nu} \right).
\end{aligned}
\tag{30}
$$

We now evaluate the diagonal and non-diagonal parts separately. We find for the diagonal contribution:

$$
\begin{aligned}
f_{nn}^{\mu\nu} &= |c_n|^2 \left(1 - \delta_{\mu_n,0}\right)\left(1 - \delta_{\nu_n,0}\right) \cdot \\
&\quad \int d\Omega \, s_1^{\mu_1} s_1^{\nu_1} \cdots \sum_{i=x,y,z} \left(\delta_{\mu_n,i} - s_n^{\mu_n} s_n^i\right)\left(\delta_{\nu_n,i} - s_n^{\nu_n} s_n^i\right) \ldots s_N^{\mu_N} s_N^{\nu_N} \\
&= 2^{N+1} \left(1 - \delta_{\mu_n,0}\right) \delta_{\mu,\nu} |c_n|^2,
\end{aligned}
\tag{31}
$$

which follows from $\int d\Omega \, s_n^{\mu_n} s_n^{\nu_n} = \text{Tr}(\hat\sigma_n^{\mu_n} \hat\sigma_n^{\nu_n}) = 2\delta_{\mu_n,\nu_n}$ and $|s_n|^2 = 3$. The off-diagonal components all vanish

$$
f_{mn}^{\mu\nu} = 0, \qquad \text{for} \quad m \neq n,
$$

as each contains factors

$$
\int d\Omega_n \, s_n^{\mu_n} \left(1 - \delta_{\nu_n,0}\right)\left(\delta_{\nu_n,i} - s_n^{\nu_n} s_n^i\right) = 0.
$$

Since

$$
\text{Tr}(\hat\rho^2) = \sum_{\mu,\nu} \rho_\mu^* \rho_\nu \text{Tr}(\hat\sigma_1^{\mu_1} \hat\sigma_1^{\nu_1}) \ldots \text{Tr}(\hat\sigma_N^{\mu_N} \hat\sigma_N^{\nu_N}) = 2^N \sum_\mu |\rho_\mu|^2,
\tag{32}
$$

we see that

$$
\begin{aligned}
||\delta(\Omega)||^2 &= 2^N \sum_\mu |\rho_\mu|^2 \sum_{n=1}^N 2|c_n|^2 \left(1 - \delta_{\mu_n,0}\right) \\
&\leq 2|\{c_n\}|^2 \text{Tr}\left(\hat\rho^2\right).
\end{aligned}
\tag{33}
$$

When $\hat\rho$ is the completely mixed state, we have $\mu_n = 0$ for every $n$ and therefore the truncated correspondence rules are exact. For general states we can infer the error to scale as

$$
||\delta(\Omega)|| \sim |\{c_n\}| = \left[ \sum_{n=1}^N |c_n|^2 \right]^{1/2}.
\tag{34}
$$

One recognizes that if the coefficients $c_n$ all have comparable magnitudes we have

$$
||\delta(\Omega)|| \sim \mathcal{O}(\sqrt{N}).
\tag{35}
$$

A necessary condition for the asymptotic correspondence rules to be valid is that the relative deviation to the exact Weyl symbol is small, i.e.

$$
\frac{||\delta(\Omega)||}{||W_{\hat{S}(\{c_n\})\hat\rho}||} \ll 1.
\tag{36}
$$

Note that

$$||W_{\hat{S}(\{c_n\})\hat{\rho}}|| = \sqrt{\text{Tr}\left(\hat{\rho}^2 \hat{S}(\{c_n\})^\dagger \hat{S}(\{c_n\})\right)}. \tag{37}$$

This expression can maximally scale as $\mathcal{O}(N)$, in which case the truncation approximation Eq. (36) is satisfied for large ensemble sizes $N$. We now argue that this is the case if the dynamics of the system takes place in the subspace of states with large cooperativity. To this end consider the totally symmetric operators with $c_n \equiv 1$. The total angular momentum operator $\hat{S}^2 = \hat{S}^\dagger \hat{S}$ has eigenstates $|j, m, \alpha\rangle$ with so-called cooperativity $0 \leq j \leq \frac{N}{2}$, projection $|m| \leq j$ on the $z$-axis and the parameter $\alpha$ distinguishing degenerate states. If $\hat{\rho} = |j, m, \alpha\rangle\langle j, m, \alpha|$ and $j \neq 0$, then Eq. (36) yields

$$\frac{||\delta(\Omega)||}{||W_{\hat{S}(\{c_n\})\hat{\rho}}||} \leq \frac{\sqrt{2N \langle j, m, \alpha | j, m, \alpha\rangle}}{|\langle j, m, \alpha | \hat{S}^2 | j, m, \alpha\rangle|^{1/2}} = \sqrt{\frac{2N}{j(j+1)}} = \mathcal{O}\left(\frac{1}{\sqrt{N}}\right), \tag{38}$$

i.e. the cooperativity of the spin ensemble determines the validity of the asymptotic form of the correspondence rules.

## 3.3 Two-body interactions and collective dephasing

Before turning to specific applications of our TWA approach, let us discuss two special cases of collective spin-spin interactions and collective dissipative processes which are relevant e.g. for ensembles of two-level atoms coupled via a cavity field.

To describe the time evolution under the action of a collective interaction we can use the truncated correspondence rules of Eq. (22) resulting in

$$-\frac{i}{2}\left[\hat{S}^x(\{c_n\})\hat{S}^x(\{c_n\}), \hat{\rho}\right] \overset{\approx}{\longleftrightarrow} \sum_{mn} c_m c_n \Big( +\nabla_{\theta_m} \sin\phi_m s_n^x + \nabla_{\phi_m} \cot\theta_m \cos\phi_m s_n^x$$
$$+ \nabla_{\theta_n} \sin\phi_n s_m^x + \nabla_{\phi_n} \cot\theta_n \cos\phi_n s_m^x \Big) W_{\hat{\rho}}(\Omega), \tag{39a}$$

$$-\frac{i}{2}\left[\hat{S}^y(\{c_n\})\hat{S}^y(\{c_n\}), \hat{\rho}\right] \overset{\approx}{\longleftrightarrow} \sum_{mn} c_m c_n \Big( -\nabla_{\theta_m} \cos\phi_m s_n^y + \nabla_{\phi_m} \cot\theta_m \sin\phi_m s_n^y$$
$$- \nabla_{\theta_n} \cos\phi_n s_m^y + \nabla_{\phi_n} \cot\theta_n \sin\phi_n s_m^y \Big) W_{\hat{\rho}}(\Omega), \tag{39b}$$

$$-\frac{i}{2}\left[\hat{S}^z(\{c_n\})\hat{S}^z(\{c_n\}), \hat{\rho}\right] \overset{\approx}{\longleftrightarrow} \sum_{mn} c_m c_n \Big( -\nabla_{\phi_m} s_n^z - \nabla_{\phi_n} s_m^z \Big) W_{\hat{\rho}}(\Omega). \tag{39c}$$

The equivalent stochastic differential equations in $\theta_n, \phi_n$ are in fact deterministic and quantum fluctuations enter only through the averaging over the Wigner distribution of the initial state. A change of variables $\theta_n, \phi_n \to s_n$ to Cartesian coordinates then gives equations of the type

$$d s_n = 2 c_n S^\mu(\{c_n\}) \times s_n dt, \tag{40}$$

where $S^\mu(\{c_n\}) = \sum_m c_m s_m^\mu e_\mu$ with $\mu = x, y, z$. This is a Larmor precession of the vectors $s_n$ about the cumulative magnetic field $2 c_n S^\mu(\{c_n\})$ and is equivalently predicted by a mean-field approximation and the standard DTWA.

In addition to unitary interactions described by a von Neumann equation, collective dissipative processes described by Linblad master equations are oftentimes of interest as well. A

particularly simple case is that of *collective dephasing* for which we find an exact mapping to a FPE [34]

$$\frac{\gamma}{2}\left[2\hat{S}^z(\{c_n\})\hat{\rho}\hat{S}^z(\{c_n\}) - \hat{S}^z(\{c_n\})\hat{S}^z(\{c_n\})\hat{\rho} - \hat{\rho}\hat{S}^z(\{c_n\})\hat{S}^z(\{c_n\})\right]$$

$$\longleftrightarrow \frac{\gamma}{2}\sum_{mn}\nabla_{\phi_m}\nabla_{\phi_n}4c_m c_n W_{\hat{\rho}}(\mathbf{\Omega}). \qquad (41)$$

This equation has the equivalent set of very simple SDEs

$$d\theta_n = 0, \quad d\phi_n = 2\sqrt{\gamma}c_n dW. \qquad (42a)$$

It is not surprising that all angles $\phi_n$ couple to the same noise $dW$, as their time evolution can equivalently be generated by a dynamic Hamiltonian contribution $\hat{H} = \gamma\hat{S}^z(\{c_n\})\eta(t)$ where $\eta(t)$ is a white noise process with identical properties as $dW$.

# 4 TWA Description of Collective Light Emission

Let us consider $N$ two-level atoms with arbitrary but non-overlapping positions $\mathbf{r}_n$ coupled to the quantized electromagnetic field at distances comparable to the wavelength $\lambda_e$ of the two-level transition. In contrast to the case of atoms spaced at distances much larger than $\lambda_e$, which allows a formal elimination of the coupling to the radiation field for each atom individually, leading to the effective Lindblad master equation (14), here radiative couplings between the atoms need to be taken into account. In addition we allow for a driving of the atoms by an external coherent light field

$$\mathcal{E}(\mathbf{r},t) = \mathbf{e}_c \mathcal{E}(\mathbf{r})e^{-i\omega_c t} + \text{c.c.} \qquad (43)$$

which is polarized along the unit vector $\mathbf{e}_c$ and has the wave vector $\mathbf{k}_c = \mathbf{e}_n \omega_c/c$ with $\mathbf{e}_n \cdot \mathbf{e}_c = 0$. The corresponding Rabi frequency for the $j$'th atom is $\Omega_j = \mathbf{p} \cdot \mathbf{e}_c \mathcal{E}(\mathbf{r}_j)$, where $\mathbf{p}$ is the atomic transition dipole moment, which is assumed to be identical for all atoms. We denote the detuning between the classical field and the atoms as $\Delta = \omega_c - \omega_e$. Formally integrating out the electromagnetic field and using a Born-Markov approximation results in a master equation of the $N$ atom system which reads [45]

$$\frac{d}{dt}\hat{\rho} = -i\left[\hat{H},\hat{\rho}\right] + \frac{1}{2}\sum_{mn}\Gamma_{mn}(2\hat{\sigma}_m^-\hat{\rho}\hat{\sigma}_n^+ - \hat{\sigma}_m^+\hat{\sigma}_n^-\hat{\rho} - \hat{\rho}\hat{\sigma}_m^+\hat{\sigma}_n^-). \qquad (44)$$

The effective Hamiltonian

$$\hat{H} = -\frac{\Delta}{2}\sum_n \hat{\sigma}_n^z - \sum_n\left(\Omega_n \hat{\sigma}_n^+ + \text{h.a.}\right) + \sum_n\sum_{m\neq n}J_{mn}\hat{\sigma}_m^+\hat{\sigma}_n^-, \qquad (45)$$

describes the coupling to the external coherent drive as well as the radiative coupling between the two-level atoms with rates $J_{mn}$. These rates as well as the positive definite decay matrix $\mathbf{\Gamma} = \mathbf{G}\mathbf{G}^T \in \mathbb{R}^{N\times N}$, where the choice of $\mathbf{G}$ is unique up to a unitary rotation, are given by the free space Green's tensor $\mathbf{G}_E(\mathbf{r}_m,\mathbf{r}_n,\omega_e)$ of the electric field

$$-J_{mn} + \frac{i}{2}\Gamma_{mn} = \frac{1}{\epsilon_0}\left(\frac{2\pi\omega_e}{c}\right)^2 \mathbf{p}^\dagger \cdot \mathbf{G}_E(\mathbf{r}_m,\mathbf{r}_n,\omega_e) \cdot \mathbf{p}. \qquad (46)$$

Their explicit expressions are

$$\frac{J_{mn}}{\Gamma_0} = -\frac{3}{4}\Bigg\{ \left(1 - |\boldsymbol{e}_p \cdot \boldsymbol{e}_{r_{mn}}|^2\right) \frac{\cos(k_e r_{mn})}{k_e r_{mn}}$$
$$- \left(1 - 3|\boldsymbol{e}_p \cdot \boldsymbol{e}_{r_{mn}}|^2\right) \left[ \frac{\sin(k_e r_{mn})}{(k_e r_{mn})^2} + \frac{\cos(k_e r_{mn})}{(k_e r_{mn})^3} \right] \Bigg\}, \tag{47a}$$

$$\frac{\Gamma_{mn}}{\Gamma_0} = \frac{3}{2}\Bigg\{ \left(1 - |\boldsymbol{e}_p \cdot \boldsymbol{e}_{r_{mn}}|^2\right) \frac{\sin(k_e r_{mn})}{k_e r_{mn}}$$
$$+ \left(1 - 3|\boldsymbol{e}_p \cdot \boldsymbol{e}_{r_{mn}}|^2\right) \left[ \frac{\cos(k_e r_{mn})}{(k_e r_{mn})^2} - \frac{\sin(k_e r_{mn})}{(k_e r_{mn})^3} \right] \Bigg\}, \tag{47b}$$

where $\boldsymbol{r}_{mn} = \boldsymbol{r}_m - \boldsymbol{r}_n$ and $\boldsymbol{e}_p$ ($\boldsymbol{e}_{r_{mn}}$) is the unit vector along the polarization $\boldsymbol{p}$ (the position $\boldsymbol{r}_{mn}$) and $k_e = \omega_e/c$. The diagonal elements $\Gamma_{nn} = \Gamma_0$ are given by the Einstein A coefficient and we set $J_{nn} = 0$, thereby absorbing it into the atomic detuning $\Delta$. This contribution corresponds to the Lamb shift, which is however not correctly described within the dipole approximation of the atom-light coupling. In fact $J_{nn}$ diverges since $r_{mn} \to 0$ for $m = n$. In the following sections we will assume resonant driving of the atoms and therefore set $\Delta = 0$.

An exact mapping of the master equation to phase space would go beyond a Fokker-Planck description, however the asymptotic correspondence rules of Eqs. (22) reduce it to

$$\frac{\partial}{\partial t} W_{\hat{\rho}}(\boldsymbol{\Omega}, t) = \left( -\mathcal{L}_1 + \frac{1}{2}\mathcal{L}_2 \right) W_{\hat{\rho}}(\boldsymbol{\Omega}, t), \tag{48a}$$

$$\mathcal{L}_1 = \sum_{n=1}^{N} \Bigg\{ \nabla_{\theta_n}\Bigg[ \frac{\Gamma_{nn}}{2}\cot\theta_n + \sqrt{3}\sum_{m=1}^{N}\sin\theta_m\left( J_{mn}\sin\phi_{mn} + \frac{\Gamma_{mn}}{2}\cos\phi_{mn} \right) \Bigg]$$
$$+ \nabla_{\phi_n}\sqrt{3}\cot\theta_n\sum_{m=1}^{N}\sin\theta_m\left( -J_{mn}\cos\phi_{mn} + \frac{\Gamma_{mn}}{2}\sin\phi_{mn} \right) \Bigg\}, \tag{48b}$$

$$\mathcal{L}_2 = \sum_{m,n=1}^{N} \Gamma_{mn}\Bigg( \nabla_{\theta_m}\nabla_{\theta_n}\cos\phi_{mn} + \nabla_{\phi_m}\nabla_{\phi_n}\cot\theta_m\cot\theta_n\cos\phi_{mn}$$
$$- \nabla_{\theta_m}\nabla_{\phi_n}\cot\theta_n\sin\phi_{mn} + \nabla_{\phi_m}\nabla_{\theta_n}\cot\theta_m\sin\phi_{mn} \Bigg), \tag{48c}$$

where $\phi_{mn} = \phi_m - \phi_n$. Note that applying the approximate correspondence rules to terms such as $\hat{S}(\{c_n\})^- \hat{\rho}\hat{S}(\{c_n\})^-$ from either left-to-right or right-to-left produces a small imaginary contribution even though the master equation is real-valued. This produces a neglectable error but can be alleviated by taking the symmetric average of both variations which corresponds to just taking the real part of either one. The above FPE possesses an equivalent set of SDEs given by

$$d\theta_n = \Bigg[ \frac{\Gamma_{nn}}{2}\cot\theta_n + \sqrt{3}\sum_{m=1}^{N}\sin\theta_m\left( J_{mn}\sin\phi_{mn} + \frac{\Gamma_{mn}}{2}\cos\phi_{mn} \right) \Bigg]dt$$
$$+ \sum_{m=1}^{N} G_{nm}(-\cos\phi_n dW_{\theta_m} + \sin\phi_n dW_{\phi_m}), \tag{49a}$$

$$d\phi_n = \sqrt{3}\cot\theta_n\sum_{m=1}^{N}\sin\theta_m\left( -J_{mn}\cos\phi_{mn} + \frac{\Gamma_{mn}}{2}\sin\phi_{mn} \right)dt$$
$$+ \sum_{m=1}^{N} G_{nm}\cot\theta_n(\sin\phi_n dW_{\theta_m} + \cos\phi_n dW_{\phi_m}), \tag{49b}$$

with $2N$ independent Wiener increments.

Note that Eq. (48) does not reduce to Eq. (15) when taking the limit $N = 1$. The latter is exact due to a representation in terms of a suitable choice of operators in the single-spin Hilbert space whereas the former uses an expansion in the total angular momentum of the ensemble of spins and therefore becomes invalid at small $N$.

The single-particle terms can be treated exactly and yield

$$\frac{i\Delta}{2}[\hat{\sigma}_n^z, \hat{\rho}] \longleftrightarrow \Delta \cdot \nabla_{\phi_n} W_{\hat{\rho}}(\mathbf{\Omega}), \tag{50a}$$

$$\frac{i}{2}[\Omega_n \hat{\sigma}_n^+ + \Omega_n^* \hat{\sigma}_n^-, \hat{\rho}] \longleftrightarrow -\left[\nabla_{\theta_n} \text{Im}\left(\Omega_n e^{i\phi_n}\right) \right.$$

$$\left. + \nabla_{\phi_n} \text{Re}\left(\Omega_n e^{i\phi_n}\right) \cot \theta_n\right] W_{\hat{\rho}}(\mathbf{\Omega}), \tag{50b}$$

which gives the following additional deterministic contributions

$$d\theta_n = \text{Im}\left(\Omega_n e^{i\phi_n}\right) dt, \tag{51a}$$

$$d\phi_n = \left[\text{Re}\left(\Omega_n e^{i\phi_n}\right) \cot \theta_n - \Delta\right] dt, \tag{51b}$$

to the above SDEs.

In the following sections we will investigate specific examples of atomic matter coupled to quantized light fields and demonstrate the strengths and weaknesses of the TWA by comparing its predictions of several observables to numerically exact results. An experimentally available observable is for example the total photon emission rate

$$I(t) = -\frac{d}{dt}\langle \hat{S}^z \rangle|_\Gamma = \frac{1}{2} \sum_{m,n=1}^N \Gamma_{mn}\langle 2\hat{\sigma}_m^+ \hat{S}^z \hat{\sigma}_n^- - \hat{\sigma}_m^+ \hat{\sigma}_n^- \hat{S}^z - \hat{S}^z \hat{\sigma}_m^+ \hat{\sigma}_n^- \rangle \tag{52}$$

into all spatial directions. It is typically easier to detect the intensity of the emitted light into a solid angle with a direction defined by the unit vector $\mathbf{e}_k$ or small areas obtained from an integration over some geometric configuration thereof. The photon emission rate along the unit vector $\mathbf{e}_k$ is given by [45]

$$I_{\mathbf{e}_k}(t) = I_{\mathbf{e}_k}^0 \sum_{m,n=1}^N e^{\frac{2\pi i}{\lambda_e} \mathbf{e}_k(\mathbf{r}_m - \mathbf{r}_n)} \langle \hat{\sigma}_m^+ \hat{\sigma}_n^- \rangle, \tag{53a}$$

$$I_{\mathbf{e}_k}^0 = \Gamma_0 (1 - |\mathbf{e}_p \cdot \mathbf{e}_k|^2), \tag{53b}$$

where $I_{\mathbf{e}_k}^0$ is the enveloping emission profile of a single atom. Note that this can be rewritten in terms of collective operators

$$I_{\mathbf{e}_k}(t) = I_{\mathbf{e}_k}^0 \langle \hat{S}(\{e^{-\frac{2\pi i}{\lambda_e} \mathbf{e}_k \mathbf{r}_n}\})^\dagger \hat{S}(\{e^{-\frac{2\pi i}{\lambda_e} \mathbf{e}_k \mathbf{r}_n}\}) \rangle \tag{54}$$

such that the Weyl symbol

$$\hat{S}(\{c_n\})^\dagger \hat{S}(\{c_n\}) \longleftrightarrow |S^-(\{c_n\})|^2 + \frac{1}{2} S^z(\{|c_n|^2\}) \tag{55}$$

required for the calculation of the expectation value follows from applying the truncated correspondence rules from right to left. The aforementioned total radiated intensity is thus explicitly given as

$$I(t) = -\frac{\sqrt{3}}{2} \Gamma_0 \sum_n \overline{\cos \theta_n} - \frac{3}{4} \sum_{mn} \Gamma_{mn} \overline{\sin \theta_m \sin \theta_n \cos(\phi_m - \phi_n)}. \tag{56}$$

Unlike cumulant expansions, which only allow the investigation of expectation values up to the truncation order, the TWA can in principle be used to calculate arbitrary expectation values. A prime example for this is the second order correlation function the detected light

$$g_{e_k}^{(2)}(\tau = 0; t) = \left| \frac{I_{e_k}^0}{I_{e_k}(t)} \right|^2 \langle \hat{S}(\{e^{-\frac{2\pi i}{\lambda_e} e_k r_n}\})^\dagger \hat{S}(\{e^{-\frac{2\pi i}{\lambda_e} e_k r_n}\})^\dagger \hat{S}(\{e^{-\frac{2\pi i}{\lambda_e} e_k r_n}\}) \hat{S}(\{e^{-\frac{2\pi i}{\lambda_e} e_k r_n}\}) \rangle, \quad (57)$$

which similarly evaluates to

$$\hat{S}(\{c_n\})^\dagger \hat{S}(\{c_n\})^\dagger \hat{S}(\{c_n\}) \hat{S}(\{c_n\}) \leftrightarrow$$
$$|S^-(\{c_n\})|^4 + 2S^z(\{|c_n|^2\})|S^-(\{c_n\})|^2 + \frac{1}{2}|S^z(\{|c_n|^2\})|^2 - S^-(\{|c_n|^2 c_n\})^* S^-(\{c_n\}). \quad (58)$$

Just like in Eq. (48) the truncation produces a small imaginary component for the Weyl symbol if complex $c_n$ are considered. This is again alleviated by taking the symmetric average over the left-to-right and right-to-left application of the correspondence rules which effectively produces just the real part of the above symbol.

Moreover we consider the spin squeezing parameter $\xi^2$ defined as

$$\xi^2 = \frac{N}{|\langle \hat{\boldsymbol{S}} \rangle|^2} \min_{e_n} \left( \Delta \hat{S}_{e_n} \right)^2, \quad (59)$$

where $\hat{\boldsymbol{S}} = \left( \hat{S}^x, \hat{S}^y, \hat{S}^z \right)^T$ is the collective spin operator and $\Delta \hat{S}_{e_n} = \langle \hat{S}_{e_n}^2 \rangle - \langle \hat{S}_{e_n} \rangle^2$ is the variance of the operator $\hat{S}_{e_n} = e_n \cdot \hat{\boldsymbol{S}}$ projected onto an axis that is orthogonal to the mean spin, i.e. $\langle \hat{\boldsymbol{S}} \rangle \cdot e_n = 0$. This minimal variance is not only of interest in quantum metrology, but furthermore a squeezing of $\xi^2 < 1$ implies entanglement [46].

## 5 Dicke Decay

To benchmark our method and to illustrate its strengths, we will first study the case where all atoms couple with identical rates $\Gamma_{mn} = \Gamma_0$, and where the unitary couplings are ignored $J_{mn} \equiv 0$. This model was proposed by Dicke as an approximation to the radiative coupling of a free gas at very strong confinement [1]. The model also typically arises in cavity- and waveguide QED.

We fix the non-unique choice of $G$ in $\boldsymbol{\Gamma} = \boldsymbol{G}\boldsymbol{G}^T$ to $G_{mn} = \sqrt{\Gamma_0}\delta_{n,1}$. Substituting this into Eqs. (49) reveals that the phase space angles only couple to 2 of the possible $2N$ white noise processes. This is intuitive, e.g. from cavity QED, where these two degrees of freedom represent the noisy coherent amplitude $d\alpha = dW_x + idW_y$ of the bosonic cavity mode that adiabatically follows the state of the atoms. If the system is initially in the inverted state $|e_1 e_2 \dots e_N\rangle = |j = N/2, m = N/2\rangle$, it can only decay along the states of maximal cooperativity $j = N/2$. Hence we expect the TWA be a good approximation at large $N$. The ensemble descends this ladder of states with initially increasing and then decreasing rates. This gives rise to the effect of superradiance, i.e. the emission of light at a rate faster than that of a single atom [1]. Furthermore the restriction to just $N + 1$ states means that an exact and efficient numerical integration of the master equation in terms of rates is possible [3].

In Fig. 1 we compare the TWA prediction of the number of excitations and the total emission rate to exact results. The TWA results were produced using an Euler-Maruyama integration scheme [42] with a timestep $\ln(N)\Gamma_0 \Delta t / N = 10^{-3}$ and an averaging over $64 \cdot 10^3$ trajectories. They accurately reproduce the exact results. Even at small ensemble sizes of

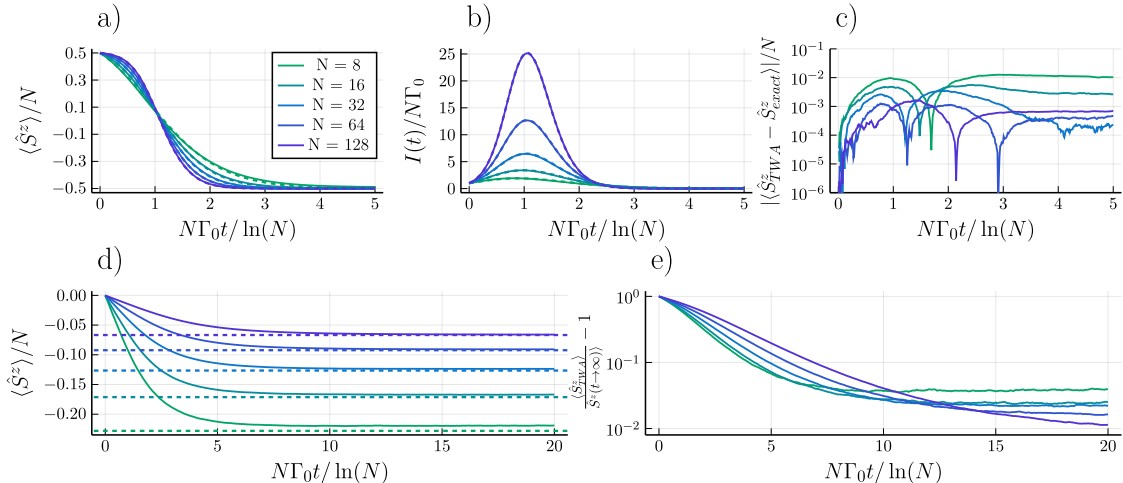

Figure 1: Dynamics of the Dicke decay for varying ensemble sizes $N$. For an initially inverted system the a) number of excitations and b) emission rate of are shown as predicted by the TWA (solid lines) and exact results (dashed lines). c) Absolute difference of the exact excitation number and TWA prediction. For the initially fully mixed state d) depicts population trapping of the excitation number. The dashed lines indicate the exact steady state populations. e) Time evolution of the relative deviation of the TWA population prediction to the exact steady state.

$N = 8$ a maximum absolute error of only $\approx 1\%$ occurs which further decreases in $N$. The positions and heights of the superradiant bursts are matched with similar accuracy.

Since only the states of maximal $j = N/2$ couple to the vacuum state $|g_1 g_2 \dots g_N\rangle$, other initial states cannot fully emit their excitations. This gives rise to the effect of *excitation trapping*. For simplicity consider even $N$. If we assume the initial state to be the completely mixed state, given by the factorized Wigner function $W_{\hat\rho}(\boldsymbol\Omega) = \prod_{n=1}^{N} W_{\hat\rho}^{(n)}(\Omega_n)$ with $W_{\hat\rho}^{(n)}(\Omega_n) = 1/2$, the steady state population can be determined by summing over the $(2j+1)d_j$ states in each $j$-ladder with degeneracy $d_j$ and with probability $2^{-N}$ each and multiplying by the population $-j$ of the bottom state, leading to

$$\langle \hat{S}^z(t \to \infty) \rangle = \sum_{j=0}^{N/2} \frac{(2j+1)d_j}{2^N}(-j), \tag{60a}$$

$$d_j = (2j+1)\frac{N!}{(N/2+j+1)!(N/2-j)!}. \tag{60b}$$

In Fig. 1 d) and e) we again see a very good agreement of the TWA with the exact results that improves as $N$ increases.

Furthermore, the Dicke decay is a prime example for revealing how superradiance emerges within a semiclassical framework. With the assumption that $J_{mn} = 0$ and $\Gamma_{mn} = \Gamma_0$ we can see that the SDEs of Eqs. (49) are closely related to the Kuramoto model [47]

$$\frac{d}{dt}\phi_n = \omega_n + \sum_{m=1}^{N} K_{mn} \sin(\phi_m - \phi_n), \tag{61}$$

which describes harmonic oscillators with frequencies $\omega_n$ and pairwise coupling rates $K_{mn}$. If we compare this to the equations of the relative phases $\phi_n$ of the two-level states, we can identify $\omega_n = -\Delta = 0$ and $K_{mn} = \frac{1}{2}\Gamma_0 \sin\theta_m \cot\theta_n$. The coupling is long-ranged and, due to the

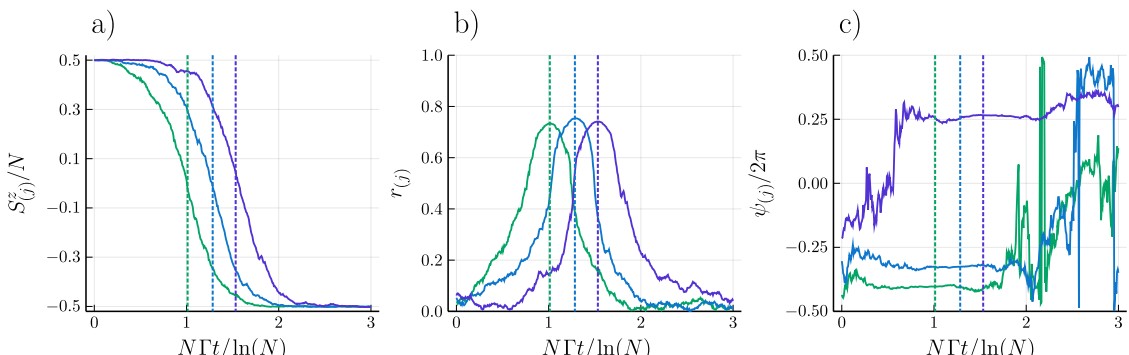

Figure 2: Time evolution of three sample trajectories according to the Dicke decay in TWA with $N = 256$ atoms. Depicted are a) the number of excitations, b) coherences and c) average phases of the atomic ensemble. The dashed vertical lines denote times of peak coherence.

appearing $\theta_m$ terms, time-dependent. Additionally, the phases are subjected to non-diagonal and non-linear noise. Nevertheless the origin of superradiant bursts in the Dicke model is related to the phase transition in the Kuramoto model from a completely incoherent state where all $\{\phi_n\}$ are uniformly distributed to that of spontaneous synchronization. This emergence of synchronization $\phi_m = \phi_n$ causes a dynamic shift of the changes $d\theta_n$ and therefore of the total number of excitation $\langle \hat{S}^z \rangle$. As a result, the photon emission rate will transition from individually radiating atoms $\gamma(t = 0) \sim N$ to collectively enhanced emission $\sim N^2$.

In the Kuramoto model, synchronization is quantified by the order parameters

$$r e^{i\psi} = \frac{1}{N} \sum_{n=1}^{N} e^{i\phi_n}, \tag{62}$$

where $0 \leq r \leq 1$ is the coherence and $\psi$ is the average phase. Individual trajectories of the Dicke decay in TWA, denoted by the subscript $(j)$, indeed share the feature of emerging transient coherence as is shown in Fig. 2. Even though the coherences $r_{(j)}$ approach zero at short and long times, there is an intermediate window where they peak significantly. Around this peak, the change of the phases in time vanishes and the signal-to-noise ratio is strongly enhanced. At the same time, the slope of the number of excitations is minimal, i.e. a photon emission burst occurs.

At first glance this coherent locking of phases might be surprising when compared to the rate equation of the density operator which does not show such an effect. We note however that the emerging average phase $\psi_{(j)}$ of a single trajectory during the burst is uniformly distributed. By taking an additional trajectory average *before* computing the order parameters, the coherence vanishes at all times.

## 6 Dynamics of Spatially Extended Systems

Let us now turn to the more realistic spatially extended systems, where idealizations such as the Dicke decay are no longer sufficient. First, we consider the recently developed light-matter interfaces based on regular arrays of atoms [48] with sub-wavelength lattice constants. In these arrays interference from the precisely positioned atomic emitters leads to pronounced collective responses despite a comparatively small number of atoms. Using atomic configura-

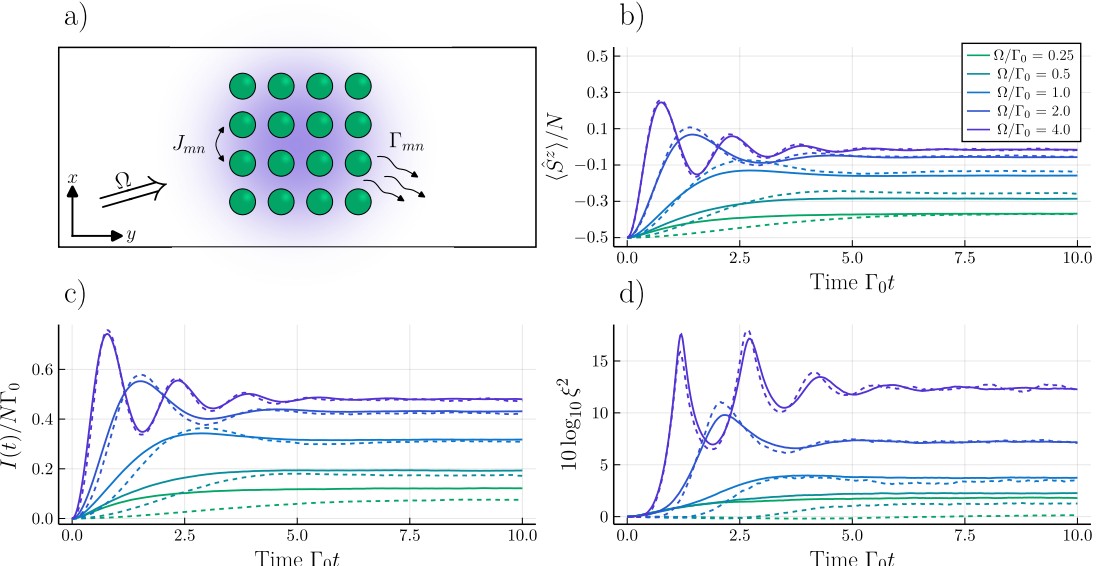

Figure 3: Dynamics of a coherently driven $4 \times 4$ quadratic atomic array with lattice spacing $a = 0.8\lambda_e$. a) Sketch of the array that is aligned in the x-y-plane and driven by a plane wave propagating along the z-axis. b) Total excitation number, c) total photon emission rate into free space and d) spin squeezing for a coherently driven atomic array as a function of time. The TWA predictions (solid lines) are compared to MCWF simulations (dashed lines). The different colors denote varying Rabi frequencies $\Omega$.

tions inspired by these recent experimental advancements, we benchmark the performance of the TWA based on the truncated correspondence rules with numerically exact results.

The atomic ensembles are treated according to their full master equation of Eq. (44) including dissipation and the dipole-dipole interactions. The numerical predictions are obtained from solving the SDEs of Eqs. (49). For all examples we choose a timestep $\Gamma_0 \Delta t = 10^{-3}$ and $N_{\text{Traj}} = 64 \cdot 10^3$ trajectories for the TWA simulations. We compare the semiclassical predictions to Monte Carlo wavefunction (MCWF) simulations obtained by using the *QuantumOptics.jl* package [49]. These are, apart from stochastic fluctuations due to a finite number of trajectories, exact. All MCWF expectation values were calculated from $N_{\text{Traj}} = 10^3$ trajectories.

Finally, we consider a dense elongated cloud of harmonically trapped atoms driven by a laser. We model the geometry and coherent drive after a recent experimental investigation [50] and study the coherence of the light emitted by the cloud.

## 6.1 Driven atomic arrays

Let us first consider an array of $N = 16$ atoms in a $4 \times 4$ quadratic lattice in the x-y-plane with lattice constant $a = 0.8\lambda_e$.

Here and in the next section, each atom is assumed to have a dipole allowed transition from the ground state $|g\rangle$ to the excited state $|e\rangle$ with circular polarization $\boldsymbol{e}_p = (1, i, 0,)^T / \sqrt{2}$ along the z-direction. They are initially in the collective ground state $|g_1 g_2 \ldots g_N\rangle$ and are driven by a plane wave which propagates perpendicularly to the array such that the Rabi frequencies are simply reduced to $\Omega_n = \Omega$.

In Fig. 3 we see that the interplay of driving and dissipation leads to damped Rabi oscillations in the number of excitations and finally to a non-trivial steady state. At small driving $\Omega/\Gamma_0 < 1$ the TWA does not reproduce the transient dynamics and steady state values obtained

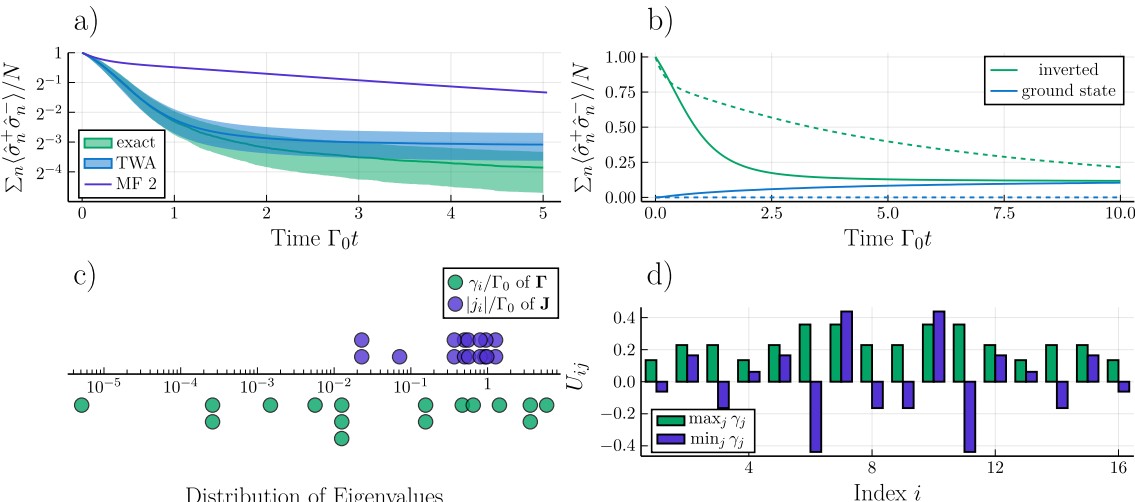

Figure 4: Dynamics of an initially inverted array at a spacing of $a = 0.2\lambda_e$ and without an external drive. a) Number of excitations as a function of time as predicted by our TWA method (blue), a second order cumulant expansion (purple, MF 2) and by an exact MCWF simulation (green). The shaded areas denote variances $\Delta S^z/2$. b) TWA (solid lines) and second order cumulant expansion (dashed lines) predictions of the number of excitations starting from the fully inverted state (green) and the collective ground state (blue). c) Distribution of the eigenvalues $\gamma_i$ of the dissipation matrix $\mathbf{\Gamma}$ and $|j_i|$ of the dipole-dipole interaction matrix $\mathbf{J}$. Stacked points denote degeneracies. d) Weights of the most superradiant and subradiant eigenvectors of $\mathbf{\Gamma}$.

from MCWF simulations, which are still feasible for this small number of emitters.

As the driving increases, the match between semiclassical and exact dynamics improves. In the moderate to strong driving regime $\Omega/\Gamma_0 \geq 1$ we see a very good agreement across all observables. Most notably, the spin squeezing parameter is matched closely, suggesting an overall excellent prediction of general second moments.

The ever improving performance in the strong driving regime $\Omega/\Gamma_0 \geq 1$ can be explained by the competition of the driving and the dissipation. Here, only the cooperative, i.e. superradiant, modes significantly contribute to the dynamics. On the other hand, subradiant modes with their weak rates become insignificant for the overall dynamics and the steady state.

## 6.2 Inverted atomic arrays

Now we analyze the relaxation of an initially fully inverted state $|e_1 e_2 \dots e_N\rangle$ in the absence of a classical driving field, i.e. we set $\Omega_0 = 0$. The atoms again form a $4 \times 4$ quadratic array in the x-y-plane with a smaller lattice constant of $a = 0.2\lambda_e$.

In Fig. 4 a) we see the comparison between the TWA prediction, second order cumulant expansion (MF 2) and a MCWF simulation. The second order cumulant expansion [20–22] result was produced using the *QuantumCumulants.jl* package. The dynamics can be split into two time regimes: The superradiant regime $\Gamma_0 t \lesssim 1$ and the subradiant regime $\Gamma_0 t \gtrsim 1$. In contrast to the second order cumulant expansion, the TWA closely matches the exact results during the superradiant burst. As the system transitions into subradiance, the semiclassical prediction converges to a finite number of excitations $\sum_n \langle \hat{\sigma}_n^+ \hat{\sigma}_n^- \rangle \approx N/10$, i.e. it *gets stuck* at a subradiant plateau which is unstable in a full quantum treatment. We verify that this is a steady state according to the TWA by comparing it to another semiclassical evolution starting from the collective ground state as shown in b) where, even in the absence of any excitation

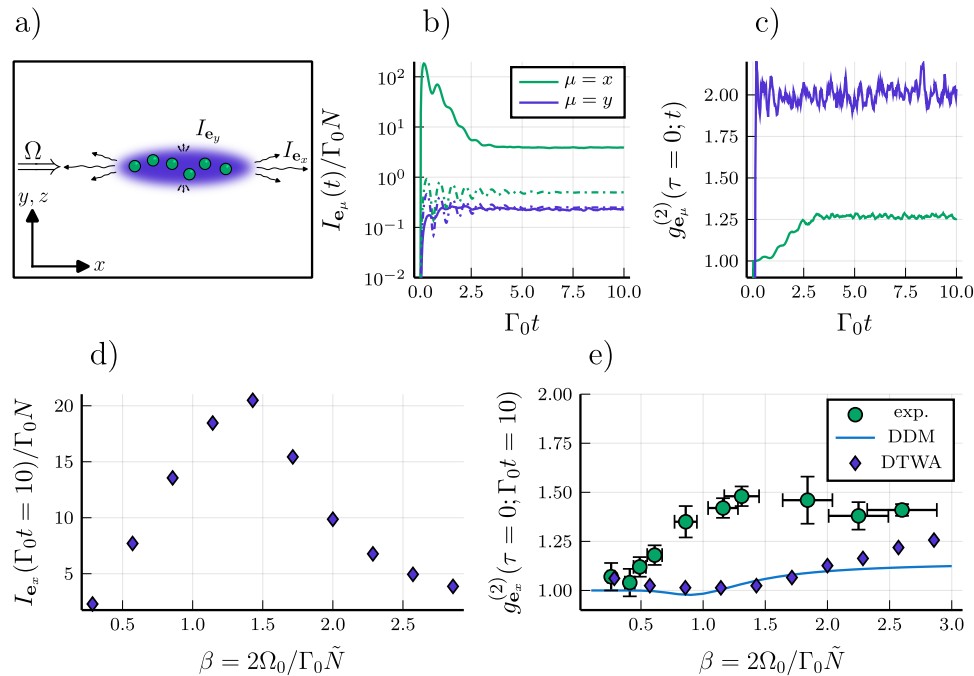

Figure 5: Dynamical and steady state radiation of a driven, cigar-shaped cloud of $N = 1400$ atoms. a) Sketch of the geometry of the driven cloud. b) Photon emission rate into the x- and y-direction and c) corresponding coherence of the emitted light at $\Omega_0 = 10\Gamma_0$ as a function of time. Steady state values of d) the photon emission rate in x-direction and e) coherences as measured (green circles), predicted by a driven Dicke model (blue solid line), both taken from [50], and the DTWA (purple diamonds) at varying $\Omega_0$.

processes, the system evolves out of the collective ground state. The second order cumulant expansion does not suffer from this problem, however fails to match the early time dynamics and similar unphysical creation of excitations in systems with subradiant modes, though not as pronounced as the TWA, has also been observed in other systems [20, 21].

This can be explained by the distribution of the eigenvalues of $\boldsymbol{\Gamma}$ and $\boldsymbol{J}$ as can be seen in c). At the timescale $\Gamma_0 t \leq 1$ the rates $\gamma_i$ of the cooperative superradiant eigenmodes of the matrix $\boldsymbol{\Gamma} = \boldsymbol{U} \cdot \mathrm{diag}(\{\gamma_j\}) \cdot \boldsymbol{U}^T$ dominate the dynamics, whereas the eigenvalues $j_i$ of the dipole-dipole interaction matrix $\boldsymbol{J}$ only significantly contribute at $\Gamma_0 t \gtrsim 1$.

In d) we show the amplitude distribution of the most superradiant (green, $j = N$) and subradiant eigenvectors (blue, $j = 1$) of $\boldsymbol{U}_{ij}$. The superradiant mode is strongly cooperative as all coefficients have the same sign and are of approximately equal magnitude. In contrast, the subradiant mode has alternating signs and couples the atoms with more varying magnitudes. The validity criterion of Eq. (36) is satisfied when the superradiant mode acts on the collective ground state, but for the subradiant mode it is not. The presence of several modes with eigenvalues $10^{-2} < \gamma_i/\Gamma_0 < 10^0$ shows that such low-cooperativity effects already become relevant at the timescale of the simulation and therefore the TWA fails to escape the plateau.

## 6.3    Superradiance from an extended atomic cloud driven by an external laser

Lastly, we consider the radiative properties of an ensemble of $N = 1400$ atoms in an extended, three-dimensional harmonic trap, see Fig. 5 a). An experimental realization of this configura-

tion was recently investigated [50]. The authors claim that the cloud of atoms behaves like an effective driven Dicke model (DDM) along the elongated x-direction, i.e. it can be described by the master equation

$$\frac{d}{dt}\hat{\rho} = -\frac{i\Omega}{2}[\hat{S}^x, \hat{\rho}] + \frac{\Gamma_0}{2}(2\hat{S}^-\hat{\rho}\hat{S}^+ - \hat{S}^+\hat{S}^-\hat{\rho} - \hat{\rho}\hat{S}^+\hat{S}^-). \tag{63}$$

However, the disordered atomic positions lead to a reduced cooperativity which is captured by an effectively smaller ensemble size $\tilde{N} = \mu N$. They found that $\mu \approx 0.005$ yields a good match between the experimental observations and the DDM.

To describe the experiment within the DTWA, the positions of the atoms are normally distributed with standard deviation $\xi = (10, 0.25, 0.25)\lambda_e$ along each dimension and vanishing mean, such that the cloud is cigar-shaped and has the reported $1/e^2$-radial widths.

The atoms are assumed to have circular polarized transitions in the y-z-plane, i.e. we choose $\boldsymbol{e}_p = (0, 1, i)^T/\sqrt{2}$. They are initially in the collective ground state $|g_1 g_2 \dots g_N\rangle$ and are are driven by a plane wave with $\boldsymbol{k}_c = \frac{2\pi}{\lambda_e}\boldsymbol{e}_z$ such that we obtain Rabi frequencies $\Omega_n = \Omega_0 e^{i\boldsymbol{k}_c \cdot \boldsymbol{r}_n}$. We perform simulations at varying driving strengths $\Omega_0/\Gamma_0 = 1, 2, \dots, 10$.

In Fig. 5 b) and c) the dynamics of the cloud at $\Omega_0 = 10\Gamma_0$, i.e. in the superradiant regime, is compared to that of a single atom in free space. The incoming field drives damped Rabi oscillations which are further suppressed due to collective effects. The photon emission rate $I_{e_y}(t)$ along the short side also shows collectively-damped oscillations, but converges to the single-particle emission rate. In stark contrast to this, the emission into the x-direction shows a strong initial superradiant burst. The steady state photon emission rate per atom $I_{e_x}(\Gamma_0 t = 10)/N$ is also greatly enhanced. The second order correlation function of the light in the y-direction immediately saturates to that of a single thermal mode, demonstrating that the atoms emit independently into this direction and that no coherent locking occurs. In contrast, photons emitted into the elongated direction show a much stronger degree of coherence.

In d) and e) we investigate the steady state radiation according to the DTWA. Here, at $\beta = 2\Omega_0/\Gamma_0\tilde{N} \approx 1.4$ the transition from the magnetic to the superradiant regime of the DDM occurs. We compare the coherence of the emitted light to the experimental results and the corresponding DDM prediction [50] and see a much closer agreement with the DDM. The higher value of $g^{(2)}$ in the experimental data suggests additional dephasing effects, e.g. due to the thermal motion of the atoms, which is not included in the DDM and DTWA descriptions.

# 7 Conclusion

We here presented a semiclassical, numerically efficient approach to describe the many-body dynamics of spins with collective interactions and dissipation. The approach is an extension of the discrete truncated Wigner approximation [24], which approximates the time evolution of a physical state in the Wigner phase space by a diffusion-like process taking into account classical and leading-order quantum fluctuations. The equation of motion of the Wigner distribution of the many-body density matrix can be cast into a differential form by applying correspondence rules, i.e. the action of an operator on the state translated into phase space. We proposed a specific truncation of said correspondence rules by only keeping lowest-order contributions which maps the Lindblad master equation of interacting two-level systems to a Fokker-Planck equation with positive diffusion. The latter allows for a numerically inexpensive propagation in time by solving only linearly many stochastic differential equations.

We derived quantifiable conditions for the validity of the approximation in terms of an upper bound on the error that is introduced by the truncation. We showed in particular that the truncation becomes exact if the many-body dynamics is dominated by degrees of freedom

with high cooperativity in a large ensemble. Thus the method is ideally suited for the analysis of collective processes such as superradiant emission of light in atomic ensembles. We benchmarked our method against exact results for the Dicke decay, which can be obtained without further approximations and found excellent agreement that improves with the number of atoms in the ensemble. In the case of small atomic arrays, we compared predictions from our semiclassical approach with exact Monte Carlo wavefunction results and showed that early superradiant timescales are well captured, however longer subradiant timescales cannot be reliably described. When the array is coherently driven with Rabi frequencies at or above the single-particle linewidth, the influence of the subradiant modes becomes negligible and the emerging dynamics is again well captured within the semiclassical approximation. Furthermore we study the dynamics of a driven, harmonically trapped, spatially extended ensemble of quantum emitters and calculated its population dynamics, direction resolved photon emission rates and their corresponding degree of coherence expressed in terms of $g^{(2)}$ correlations. The experimental detected light is more thermal than the TWA simulation and the prediction from a simpler theoretical model, which suggests the existence of dephasing mechanisms not yet present in the current theory.

Our approach paves the way for studying strongly cooperative effects in large and spatially extended ensembles of two-level systems. Specifically, recent light-matter interfaces such as trapped gases and atomic arrays and their non-linear response to incoming coherent light [48,50] can be studied with ensemble sizes much larger than $N \simeq 50$ as is considered in this work. Typically, not much analytical progress can be made in such systems and methods based on tensor networks do not work reliably due to the high dimensionality of these setups and intrinsic long-range interactions.

Future works will investigate whether the truncation, which so far is a diffusion approximation, can be improved by extending it to a jump-diffusion approximation by including classical Poissonian jump processes. This might allow for an extension of the validity of our theory to regimes of moderate or even low cooperativity. Motivated by the excellent agreement of spin squeezing and coherence of the emitted light, we believe that our method can be used to analyze the generation of non-classical states of realistic, experimentally realizable atomic configurations and their emitted light.

**Funding information** The authors gratefully acknowledge financial support by the DFG through SFB/TR 185, Project No. 277625399.

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
