# Peer review of "Collective Radiative Interactions in the Discrete Truncated Wigner Approximation"

_SciPost Physics_

## Round 2 · Referee Report · Anonymous · 2023-9-10

Strengths

1) a numerical method for simulating collective radiation phenomena for large atomic ensembles and inhomogeneous coupling
2) analytic estimates of the expected errors
3) comparison with exact numerics and honest discussion about the physics that can and cannot be accurately captured by this method.

Weaknesses

missing discussion about the relation to previous work and alternative approximation schemes

Report

Collective radiation effects play an important role in many quantum optics experiments, but apart from highly idealized setups, such effects cannot be simulated exactly. Given the current renewed interest in this topic, the approximation scheme presented in this paper is very timely and considerably extends related previous works based on the DTWA, for which -- to my understanding -- a generalization to collective decay channels was not possible. The method is described in very much detail and both analytic estimates for the approximation error and a comparison with numerically exact results for small systems are presented.
Overall, in view of the relevance of the addressed problem and the quality of the presentation, the paper is in principle suited for publication in SciPost Physics.

However, before publication, there are two points that should be clarified:

1) The stochastic equations of motion for the multi-atom case in Eq. (43) do not seem to reduce to the single atom decay process given in Eq. (14) when N=1 is considered. It would be interesting to understand the differences between the current method an the one presented in Ref [25]. This is particularly relevant for the discussion of Fig. 3, where the atoms behave almost as individual atoms.

2) Collective decay processes are often very well captured by cumulant-expansion methods. The authors mention that such methods also lead to unphysical excitations, but less pronounced than with the current method. Therefore, it would be interesting to see a direct comparison of this method with a second-order cumulant expansion simulation.

Requested changes

1) Clarify the relation between the current method and the previous single-particle decay, Eq. (14).

2) Compare the current method with cumulant expansion techniques.

3) Optional: In Eq. (19) and in the following analysis the vector of weights J=(J_1,j_2, ...) is introduced. Since in the relevant literature this symbol is very often used for the collective angular momentum operator or the total angular momentum operator J=S+L, this notation is a bit confusing. The authors might consider using a different symbol for clarity of the presentation.

---

## Round 2 · Referee Report · Anonymous · 2023-10-13

Strengths

1. Expands the scope of an existing computational technique to tackle problems related to the correlated dissipation of spin systems, for which there is a current lack of viable approaches.
2. Provides relatively simple, practical equations for obtaining efficient dynamical approximations for interacting spin-1/2 systems with correlated dissipation.
3. Analysis of error rate, and demonstration of expected validity for sufficiently cooperative systems in the large N limit.
4. Detailed comparison to exact numerics for an array of timely and physically relevant problems, and discussion of applicability/validity.

Weaknesses

1. Some minor issues related to clarity/grammar.

Report

This paper reports on a key extension of an efficient computational method based on the truncated Wigner approximation for the dynamics of open spin systems. In particular, the authors discuss the extension of their recently developed hybrid-discrete-continuous Wigner approximation — which can be used to compute the effects of individual dissipation for spin systems — to the case of correlated dissipation. The authors present error estimates for their method, and compare their results to exact methods for a range of physically relevant problems.

Such a method is highly relevant, given the current lack of suitable methods for studying quantum dynamics in the presence of correlated dissipation, which have mostly been restricted to exact techniques for small systems, short times, or system exhibiting a high degree of symmetry. Furthermore, given the revival of interest in subradiant/superradiant phenomena for quantum science applications, as well as the current interest in computational techniques based on the truncated Wigner approximation for computing quantum dynamics, the presented method is very timely.

The manuscript is generally clear and well-written. It also provides an important computational advance for tackling a difficult and relevant class of problems, and is thus in principle suited for publication in SciPost Physics, pending the following points are addressed.

Requested changes

1. I agree with the previous assessment that the notation used for the weighted spin operators is confusing, since the coefficients J do not represent the total angular momentum, but appear alongside notation for spin/angular moment operators S and L. The authors might consider altering this notation to enhance clarity.
2. The line preceding the introduction of Eq. 6 seems to be referring to the wrong equation (Eq. 5 should be referenced, not Eq. 4).
3. There are some instances of missing words/grammatical issues throughout that should be checked to enhance clarity.
4. Fig. 3c is currently not discussed in the text. There is nothing to compare to the single-particle case nor does this seem to be indicating the presence of any entanglement, so its purpose is unclear to me. The authors might consider providing some discussion of this figure in the text.

---

## Editorial Decision

resubmitted